# Laboratory Test and Geochemical Modeling of Cement Paste Degradation, in Contact with Ammonium Chloride Solution

**DOI:** 10.3390/ma15082930

**Published:** 2022-04-17

**Authors:** Barbara Słomka-Słupik, Krzysztof Labus

**Affiliations:** 1Faculty of Civil Engineering, Silesian University of Technology, 44-100 Gliwice, Poland; 2Faculty of Mining, Safety Engineering and Industrial Automation, Silesian University of Technology, 44-100 Gliwice, Poland; krzysztof.labus@polsl.pl

**Keywords:** ammonium chloride, cement paste degradation, geochemical modeling, phases changes, porosity changes, chloride concentration, CEM I

## Abstract

Concrete tanks, in coke wastewater treatment plants, are exposed to aggressive wastewater with high ammonium and chloride content, deteriorating the concrete binder. Due to this, toxic compounds may migrate to the environment. The results of the experimental work presented confirmed the changes in the phase, microstructure and concentration of chlorides caused by the penetration of NH_4_Cl into the hardened cement paste in dry conditions. Geochemical modeling of the interactions between the aggressive solution, the cement stone matrix and the pore water was performed in order to track the destruction process effects. The results are useful for condition assessment of the structures operating under occasional immersion.

## 1. Introduction

It is widely known that ammonium and chloride ions are detrimental to the microstructure of concrete. NH_4_^+^ ions are derived from agriculture and industrial processes, and damage concrete structures. Ammonium ions react with the solid phases of cement binder leading to the Ca-depletion, pH reduction and therefore to an increase in porosity [1,2,3,4,5,6,7,8,9,10,11,12,13,14,15,16,17,18]. The loss of calcium ions in the components of the solid phases of the hydrated cement and, their transfer to liquid solution in the pores of the concrete, contribute to the so-called decalcification process. During lowering of the pH of the hardened cement paste, the main reactions are dissolution of portlandite—Equation (1), transformation of AFm into Kuzel’s salt—Equation (2) transformation of Kuzel’s salt into Friedel’s salt—Equation (3) decalcification of C-S-H gel—Equation (4) and transformation of sulphate phases, ettringite into thaumasite—Equations (5) and (6) and thaumasite into gypsum—Equation (7). Chloride ions, in turn, are corrosive to steel rebars and in reaction with hydrated cement phases forms Kuzel’s or Friedel’s salt [10,19,20,21,22]. However, the penetration of an aggressive medium, and the reduction in the pH of the solution in the pores, causes the dissolution of chloride phase as well—Equation (8).

Loads in reactions do not always balance (for example Equation (7)) because cement paste is a very complex medium, thus it is difficult to cover each process in one reaction. Chloride ions diffuse from this micro-area of reaction and are transported according to charge balance law [23]. Hydrogen ions come from the hydrochloric acid formed in this micro area, and support the destruction phenomena. The pH of cement paste has a huge impact on the stability of the phases. According to various scientific sources, Table 1 has been prepared [1], with the given pH values of the liquid in the pores, in which phases are stable. Most of the cement hydrates are stable at pH 13. The notation of oxides is represented as follows: C: CaO, S: SiO_2_, A: Al_2_O_3_, F: Fe_2_O_3_, H: H_2_O, Ŝ: SO_3_, Ĉ: CO_2_, M: MgO.
Ca(OH)_2_ + 2NH_4_Cl → Ca^2+^ + 2Cl^−^ + 2H_2_O + 2NH_3_↑(1)
3CaO·Al_2_O_3_·CaSO_4_·12H_2_O + NH_4_Cl → 3CaO·Al_2_O_3_·0.5CaSO_4_·0.5CaCl_2_·11H_2_O + 0.5H_2_SO_4_ + NH_3_↑ + H_2_O(2)
3CaO·Al_2_O_3_·0.5CaSO_4_·0.5CaCl_2_·11H_2_O + NH_4_Cl → 3CaO·Al_2_O_3_·CaCl_2_·10H_2_O + 0.5H_2_SO_4_ + NH_3_↑ + H_2_O(3)
xCaO·SiO_2_·zH_2_O + mNH_4_Cl → (x−δ_1_)CaO·(1−δ_2_)SiO_2_·(z−δ_1_−m)H_2_O + δ_1_[Ca(OH)_2_] + δ_2_[SiO_2_] + mNH_3_↑+ mH_3_O^+^(4)
Ca_6_[Al(OH)_6_]_2_·(SO_4_)_3_·26H_2_O + 2HCO_3_^−^ + 2H_2_SiO_4_^2−^ ↔ Ca_6_[Si(OH)_6_]_2_·(CO_3_)_2_·(SO_4_)_2_·24H_2_O + 2Al(OH)_4_^−^ + SO_4_^2−^ + 2OH^−^(5)
Ca_6_[Al(OH)_6_]_2_· (SO_4_)_3_·26H_2_O + Ca_3_Si_2_O_7_·3H_2_O + CaCO_3_ + CO_2_ + xH_2_O + CaO + 8 NH_4_^+^ ↔ Ca_6_[Si(OH)_6_]_2_·(CO_3_)_2_·(SO_4_)_2_·24H_2_O + Al_2_O_3_·xH_2_O + CaSO_4_·2H_2_O + 7H_2_O + 4Ca^2+^ + 8NH_3_↑(6)
Ca_6_[Si(OH)_6_]_2_·(CO_3_)_2_·(SO_4_)_2_·24H_2_O + 4NH_4_Cl ↔2CaSO_4_·2H_2_O + 2H_4_SiO_4_ + 2CaCO_3_ + 26H_2_O + 2Ca^2+^ + 4Cl^−^ + 4NH_3_↑(7)
CaO·Al_2_O_3_·CaCl_2_·10H_2_O + 2NH_4_Cl → 2Al(OH)_3_ + 8H_2_O + 2Ca^2+^ + 4Cl^−^ + 2NH_3_↑(8)

The decalcification process has been modeled by many researchers who have used ammonium nitrate solution (NH_4_NO_3_) in order to investigate accelerated Ca-leaching, often under the load applied [24,25,26,27,28]. Segura et al. [29] modeled the decalcification process kinetics and the degradation grade using Fick’s second law of diffusion, and the shrinking unreacted-core model. Phung et al. [30] proposed a one-dimensional diffusion-based transport model, based on macroscopic mass balances for Ca in aqueous and solid (portlandite and C-S-H) phases, linked together by applying solid-liquid Ca equilibrium. This model also simulated the water permeability through the hardened cement paste at different immersion periods in NH_4_NO_3_ solution. Song et al. [31], in turn, prepared a leaching test of Ordinary Portland Cement CEM I hardened paste in 1 M ammonium chloride aqueous solution, which also noticed calcium leaching from the cement matrix. This phenomenon greatly changed the microstructure of the cement paste, changing the chloride diffusion behavior and lowered the chloride-ingress resistance. Researchers also confirmed that in the cement paste subjected to the coupling interaction of calcium leaching and chloride diffusion, chloride diffusion behavior is remarkably different from the results predicted by the classical Fick’s 2nd law, and due to the microstructure degradation, the chloride profile fronts in the degraded zone, move faster towards the inner parts of the sample [31]. Recently, Xiang-Nan et al. [32] described the coupled chloride and calcium diffusion in concrete. Numerical simulation, verified by the experiment of a concrete plate, immersed in different corrosion solutions, has been performed. Researchers confirmed the conclusions obtained earlier, that calcium leaching increases the concrete porosity and chloride diffusivity, and chloride attack can accelerate the leaching process of concrete. Moreover, it was confirmed that the decalcification of C-S-H gel, and decomposition of Friedel’s salts increase the free chloride ions content in concrete [32].

However, the literature does not include geochemical models of concrete degradation, specifically under the influence of ammonium chloride solution. Feng et al. [33] introduced the model of leaching based on the Thermodynamic Hydration and Microstructure Evolution (THAMES) model, which enables the simulation of cement microstructure-property relations, during chemical degradation. In this model, all phases, except the clinker minerals, are treated as a closed thermodynamic system, while the clinker minerals are taken to be outside this system, and their extent of dissolution at any time sets the chemical composition of the thermodynamic system itself. The leaching of the cement microstructure is accomplished by diluting the pore solution by extracting the calcium and alkali species. 

Earlier studies, included in work [3], included a comparison of the degradation of hydrated cement pastes made of two different cements: CEM III/A 32.5N-LH/HSR/N and CEM I 42.5N-HSR/NA (according to EN 197-1), after 4 and 25 days of aggressive immersion in NH_4_Cl liquid solution. In this paper, we present the cement paste degradation process carried out with the use of CEM I 42.5R (according to EN 197-1), after 4, 19 and 25 days of the same kind of aggressive immersion, as previously noted [3]. The choice of cement was due to the fact that many tank structures for liquid waste were made of this basic cement in the last century. Currently, such tanks are operating in the ground, for example in coke wastewater plants, and are probably leaking. The degree of concrete degradation should be recognized, but it is not possible to exclude an exemplary reinforced tank from the treatment process for sampling. Moreover, even if it was possible, the test material would already be dried, and would have a different phase composition.

The solution is to perform laboratory experiments coupled with geochemical simulations, to predict the time of destruction. For these reasons, in this work, the results of laboratory tests on the degradation of the hardened cement paste (after drying) were compared and discussed with the results of simulation of geochemical changes (in saturated conditions with aggressive solution and pore liquid) in the hardened cement paste.

The aim of the study was to present the results of the destruction of the hydrated cement paste by means of laboratory tests of dry material and by means of a geochemical model for the hardened cement paste working under immersion conditions. In this way it is shown that degradation takes other forms. Both research methods were described, porosity and phase composition tests were performed, microscopic observations of the undamaged material were performed, and the model was described. Laboratory tests of the hardened cement paste were carried out and the degradation front changes were shown after 4, 19 and 25 days of immersion in NH_4_Cl using XRD, SEM and analytical tests. Changes in hardened cement paste using a geochemical model are presented in various analytical variants. The work was summarized with a discussion of the results and conclusions.

## 2. Materials and Methods

### 2.1. Laboratory Examination

#### 2.1.1. Materials

In this research 2 kinds of cement were used: Portland cement CEM I 42.5R and Portland cement CEM I 52.5R. The chemical composition of cements is shown in the Table 2. 

Cement CEM I 52.5 R gives greater strength of the binder, and differs from CEM I 42.5 R by a slightly higher Si/Ca ratio. Portland cement CEM I 42.5 R hydrated paste, according to EN 197-1 has early strength >18 MPa, standard strength >40 MPa, initial setting time >50 min. Standard strength of CEM 52.5 R, in turn, is >52 MPa, its 28-day compressive strength = 64.8 MPa. Blaine specific surface area (SSB) of CEM 52.5 R is 5120 cm^2^/g. Hardened cement paste with w/c = 0.4 made of CEM I 52.5 R and distilled water was not subjected to destruction; it has been tested for porosity (MIP), and also underwent microscopic observation (SEM/EDS) and phase composition analysis (XRD) after 23 days of hydration. These results were input data to create the geochemical model, and the hardened specimen was called C.I.3 (Figure 1).

The hardened cement paste degradation process was carried out with the use of Portland cement CEM I 42.5 R in saturated, 27%, aqueous solution of ammonium chloride (370 g of pure for analysis NH_4_Cl with 1 L of deionized water). Corroded and virgin zones were examined for microscopic observation (SEM/EDS), phase composition (XRD), chloride ions diffusion and porosity using SEM-BSE method. Specimens used in the research were prepared at a mass water/cement ratio of 0.4. Pastes were molded in 60 × 250 × 250 mm forms for 2 weeks. After this time, the hydrated cement paste specimens were demolded and cured in saturated lime water for 3 months, then subsequently stored in saturated NH_4_Cl solution for 4, 19 or 25 days (specimens: B-4, B-19 and B-25, respectively) at a temperature of 20 ± 2 °C. To avoid a change in NH_4_Cl concentration, and to ensure saturation conditions, an excess of the ammonium salt was maintained at the bottom of the container. Reference specimen B-P was also prepared and immersed in lime water for 4 months. After the immersion period, all of the specimens were dried for 2 days in air at an ambient temperature of 20 ± 2 °C and humidity of 60%.

#### 2.1.2. Methods

Porosity examination of hardened cement paste

To carry out the porosimetric test on specimen C.I.3, the mercury AUTOPORE 9500 porosimeter (Micromeritics; USA, Norcross, GA 30093) was used, enabling the study of porosity and pore size distribution in the range from 0.006 to 450 μm. The contact angle between mercury and materials was assumed to be 140°. This test was performed on crumbs.

Porosity examination of corroded B-4, B-19, B-25 samples and B-P comparative sample, with the use of SEM and image analysis, is described in [3]. This test was performed on polished sections.

Layers sampling

Layers with thicknesses of 0.5 mm, 1.0 mm, 1.5 mm or 2.0 mm were collected as powdered material by means of a special device, equipped with a rotating diamond head. This material was taken from 250 × 250 mm surfaces of each corroded B-4, B-19, B-25 samples and from the reference sample B-P. Powdered material from each layer was collected from about 385 cm^2^, and was stored in airtight glass jars with double-seal caps (Figure 2).

Phases examination (XRD)

Sample C.I.3 preparation: quartering, micronization in hexane and subsequent homogenization by vibratory milling. The amorphous fraction of the sample was estimated. Instrument: Bruker-AXS D8 Advance (Germany Bruker Corporation, 40 Manning Road, Billerica, MA, United States of America) with 2θ/θ measurement geometry and position-sensitive LynxEye detector. Measurement conditions: CuKα Ni radiation filter—voltage 40 kV, current 40 mA, step 0.014° 2Θ with time in step 0.25 s, summation of 5 successive measurements, qualitative evaluation using the BrukerDiffracSuite program: Diffrac EVA, quantification using Rietveld analysis (Bruker Topas version 4.1).

Phase composition analyses of samples B-4, B-19, B-25 and B-P were made using an X’Pert Pro MPD X-ray diffractometer, produced by PANalytical (Westborough, MA, USA). The measurements were conducted at room temperature, using monochromatic CuKα radiation. Qualitative analysis with the support of the ICDD PDF4+ database was performed employing HighScore v4.9 software (Malvern Panalytical Ltd., Malvern, UK, 2008). The test was conducted on powdered samples from corroded specimens and the reference specimen.

SEM observations

For microscopic observation, a freshly prepared refractive area of the sample C.I.3 was used without further treatment. An FEI Quanta-650 FEG auto-emission electron microscope was used to obtain photographic documentation and point microanalysis, equipped with analyzers: energy dispersive analyzer (EDX)—EDAX Galaxy, wave dispersion analyzer (WDA)—EDAX LEXS, cathodoluminescence detector (CL)—Gatan MonoCL4 and backscattered electron diffraction detector (EBSD). Under these conditions, energy dispersive microanalysis needs to be considered as semiquantitative. The microscope operated under the following conditions: voltage 20 kV, current 8–10 nA, beam diameter 6 μm, reduced vacuum with a chamber pressure of 50 Pa, samples without plating. 

Samples B-P and corroded ones (B-4, B-19, B-25) were observed in high resolution Scanning Electron Microscope (SEM) Zeiss Supra 25 (Carl Zeiss AG, Munich, Germany), using Smart-SEM and Leo 23 software, applying backscattered electron imaging of the polished sections or secondary electron imaging of fractures.

### 2.2. Modeling

#### 2.2.1. General Information on the Performed Geochemical Simulations

Software and model used

Geochemist’s Workbench (GWB) [34] version 11.0 was used in equilibrium modeling, as well as reactive transport and reaction modeling. The X1t package was used in the simulations, assuming a specific mineralogical composition of the rock matrix and the chemical composition of the pore solution. The simulation assumes that the fluid of a given composition enters the model domain, set out in the linear coordinates, and while moving through it, reacts with the components of the system (mineral phases and pore water). In our modeling, a "flat" model of a cement sample was assumed, focusing on the analysis and forecasting of geochemical reactions and variables that control the progress and products of the reaction, than on the geometry of the flow field. 

Input data

The simulations required a number of input data describing the following aspects of model construction:-mineralogical composition of the cement sample, treated as a set of minerals that are reactants in kinetic reactions;-kinetic reaction rates and the specific surface areas of individual minerals;-chemical composition of pore waters;-sample porosity;-chemical composition of the solution saturated with NH_4_Cl).

The general assumptions allowing for the definition of the model input data are presented below.

Reaction kinetics parameters

The following kinetic dissolution/precipitation rate equation, simplified after Lasaga [35] was used in the calculations, according to which a given mineral crystallizes when it is supersaturated (or dissolves when unsaturated), at a rate dependent on the reaction rate constant and the specific surface of this mineral:(9)$$r_{n}=\pm kA_{n}{\left(1-\Omega_{n}^{\theta }\right)}^{\eta },$$
where: *r_n_*—reaction rate of mineral *n* [mol·s^−1^·g^−1^]; dissolution—*r_n_* > 0, crystallization—*r_n_* < 0), *k* is the rate constant [mol·m^−2^·s^−1^], *A_n_*—the reactive surface of the mineral [m^2^·g^−1^], Ω*_n_*—is the saturation index of the mineral *n*, The empirical parameters *θ* and *η* are positive numbers, determined from experiments or usually taken as 1.

The kinetic rate constants, Table 3, were taken from literature [36,37,38,39,40,41,42,43] and recalculated according to the temperature of the simulated environment. The Geochemist’s Workbench (GWB) software, used in reactive transport and reaction modeling calculates surface area from a specific surface area—*A_n_* (cm^2^/g) and the value of the rate constant—*k*_25_ (mol·m^−2^·s^−1^), entered by the user for each kinetic mineral. The dependence of the reaction constant on the temperature follows the Arrhenius law, and can be calculated on the basis of a formula, taking into account the neutral, acidic or basic mechanisms:(10)$$\begin{array}{l} k_{{}_{T}}\;=\;k_{N25}\;\exp\left[\frac{-E_{N}}{R}\left(\frac{1}{T}\;-\;\frac{1}{298.15}\right)\right]+k_{H25}\;\exp\left[\frac{-E_{H}}{R}\left(\frac{1}{T}\;-\;\frac{1}{298.15}\right)\right]a_{H}^{n_{H}}\\ \;\;\;\;\;\;\;+k_{OH25}\;\exp\left[\frac{-E_{OH}}{R}\left(\frac{1}{T}\;-\;\frac{1}{298.15}\right)\right]a_{OH}^{n_{OH}}, \end{array}$$
where: *k*_25_—rate constant at 25 °C—(mol·m^−2^·s^−1^); *E*—activation energy (J/mol); *R*—gas constant (8.3143 J/K·mol); *T*—absolute temperature (K); *a*—ion activity; *n*—reaction order; index: *N, H, OH*-neutral, acidic and basic mechanism, respectively.

Thermodynamic database

The thermmoddem database (Bureau de Recherches Géologiques et Minières) was used for the calculations using activity coefficients calculated according to the extended Debye-Hückel equation.

Compositions of the pore solution and a saturated solution of NH_4_Cl

The chemical composition of water used in the simulations was determined based on the equilibrium modeling between the pore solution, with a set of minerals (included in the hardened cement), for this purpose the SpecE8 simulator was used [34]. The composition of saturated NH_4_Cl solution was calculated based on the solubility of ammonium chloride salt in tap water—Table 4.

#### 2.2.2. One-Dimensional Model of Transport and Reaction

Model assumptions

To illustrate the scenarios of the propagation of changes in the cement sample, in contact with a saturated NH_4_Cl solution, the transport and reaction modeling was applied, with the use of the X1t package (GWB). It was assumed that the diffusion of the NH_4_Cl solution into the interior of the sample should cause both changes in pore water chemistry and changes in the mineralogical composition of the cement sample matrix. This means that for a reactive solution, its chemistry is related to the rate of diffusion transport and, at the same time, to the rate of chemical reactions supplementing (for example, by dissolving the rock matrix) or depleting its composition (for example, due to precipitation of mineral phases or degassing). The general formula, describing the above-mentioned relationships, for one-dimensional transport has the following form:(11)$$\frac{\partial \left(\phi C_{i}\right)}{\partial t}=\frac{\partial }{\partial x}\left(\phi D_{L}\frac{\partial C_{i}}{\partial x}\right)-\frac{\partial }{\partial x}\left(\phi \nu_{x}C_{i}\right)+\phi R_{i},$$
where: *φ*—porosity [–]; *R_i_*—reaction rate [mol/cm^−3^ s^−1^]; *C_i_*—concentration of the substance *i* in the solution [mol/cm^−3^]; *D_L_*—linear dispersion coefficient [cm^2^/s]; *ν_x_*—flow velocity [cm/s], *t*—time [s].

The heat transport was also included in the calculations, although to minimize its impact it was assumed that the NH_4_Cl solution, cement and pore fluids had the same temperature.

The model assumptions are as follows:1the cement sample is simulated in a cuboid domain, with dimensions of 50 by 10 mm by 10 mm, appropriate for tracing the phenomena occurring on a local scale. The domain consists of 25 identical cells, which allows a simulation of the transport and reactions taking place in the sample with sufficient precision; this also ensures optimal computational time. The transport of the solution is only related to its diffusion into the interior of the sample;2physical properties of the simulated medium:diffusion coefficient—1·10^−6^ cm^2^/sporosity—19.84%thermal conductivity—4·10^−3^ cal/cm·s·°C (1.675 W/m·K)longitudinal dispersion—0.1 cm;3The sample matrix consists of minerals whose reaction kinetics parameters are taken from the literature [36,37,38,39,40,41,42,43], as are the specific surface of mineral grains;4The porosity of the system results from the ratio between the volume of the solution and the sum of the volume of the solution and the rock matrix;5The experiment time is 40 days.

## 3. Results 

### 3.1. Laboratory Examination

#### 3.1.1. Examination of Not Degraded Hardened Cement Paste

XRD of C.I.3

The results of the semi-quantitative analysis of 23-days hydrated cement paste made of CEM I are given in the table together with the estimation of the random error of determination. The table also shows the parameters characterizing the accuracy of the diffraction record quantification, respectively accuracy of interpolation of diffraction record, using calculated diffraction record (R_wp_, R_exp_). The value of R_wp_ characterizes the achieved interpolation error, the value of R_exp_ is an estimation of the lowest possible error under the given conditions.

The estimated crystallinity of the sample is 80%. Thus, the sample may contain a 20% amorphous phase, which, for the modeling, was assumed to be formed by C_1_SH. Phases’ mass contents from the XRD analyses were recalculated to volume contents, considering the rock porosity, and adopted in geochemical modeling. Phase composition of the hardened cement paste is presented in the Table 5. 

The sample contains a high proportion of portlandite, perhaps because it is a short hydration time (23 days). Calcium silicates and ettringite are also relatively high. It is not possible to decide from the diffraction data, which additional cations are included in hydrocalumite (Friedel’s salt). In addition to chlorides, they can represent carbonates and probably also sulfate ions. Electron microanalysis (see below) did not confirm even trace of chloride contents in hardened paste of CEM I 52.5 R. 

Results of electron microscopy SEM and microanalyses EDS of the C.I.3 sample

The sample contained abundant particles of submicron size. These cannot be reliably analyzed. Both point microanalyses and the average composition of the analyzed areas indicate the absence of chlorides. Cl^−^ contents are below the detection limit of the method (approx. 0.1%).

Photographic documentation of the sample is given in the Figure 3, where we can clearly see the huge amount of petal portlandite and unreacted, in part (bright), calcium silicate grain (O: 33%, Si: 13%, Ca: 50% of mass weight). The results of area of EDS microanalyses are also presented in the Figure 4.

Porosity of hardened cement paste of C.I.3 before degradation

The porosity of hardened cement paste of C.I.3 before degradation is shown in Figure 5. Median pore diameter was equal to 0.0684 μm, total pore surface area—11.832 m^2^/g, tortuosity—1095.86. The value of porosity was equal to 19.84%, and this value was taken to the model in this work.

#### 3.1.2. Examination of Degraded Hardened Cement Paste

Macroscopic observations

Based on the macroscopic observations of corroded fractures, shown in Figure 6, made soon after the removal of specimens from the NH_4_Cl solution, and after 2 days of drying, the thickness of the damaged zone was initially assessed. A bright layer was detected for the areas which were in direct contact with aggressive solution. The thickness of changed layer was about 2.75 mm, 5.25 mm, and 6 mm from the external surface, in case of B-4, B-19 and B-25 specimen, respectively. Figure 6 shows a crack, situated at a depth of 5.5–6.0 mm from the surface of the B-25 sample.

Phases transformations

The notation of oxides is represented as follows: C: CaO, S: SiO_2_, A: Al_2_O_3_, F: Fe_2_O_3_, H: H_2_O, Ŝ: SO_3_, Ĉ: CO_2_, M: MgO. 

In Figure 7 a set of X-ray patterns of hardened cement paste from selected layers of the B-4, B-19 and B-25 samples and the comparative B-P sample are shown. In the comparative sample B-P, the following were identified: portlandite (1), ettringite (3), calcite (6), carboaluminate (9), brownmillerite (13), katoite, alite, belite (11) and C_3_A.

In general, the dominant phases in the B-4 inner layers were portlandite (1), alite and belite (11). However, in the 4.5 ÷ 5.0 mm layer, a high content of Friedel’s salt (2) was also found, and the intensity of its reflections decreased as the distance from the sample surface. Ettringite (3) occurs at a depth of 9.0 to 2.0 mm, and thaumasite (4) at 4.0 ÷ 2.0 mm from the surface. In this case, the thaumasite is presumably formed from a solid solution with ettringite. The most stable phase was probably calcite (6), and the remaining one was brownmillerite (13). Calcite was accompanied by vaterite (7), detected in the surface layers at a depth of up to 2.5 mm, and in 4.5 ÷ 5.0 mm from the surface. Calcium aluminates have also been identified among the carbonate phases. As carboaluminate and C_4_AH_13_ (12) began to disappear, however, it was possible to identify Friedel salts.

In the sample B-19, the portlandite disappeared, and from the layer 4.5 ÷ 5.0 mm it was no longer possible to detect it. Alite, whose peaks in the inner layers down to a depth of 4.5 mm from the sample surface, also disappeared. By contrast, belite, brownmillite, and calcite were found throughout the whole area. Vaterite (7) was present in the surface layers 0 ÷ 3.5 mm. This sample shows the order in which the sulfate phases occur, as the pH decreases. At the following distances from the surface, there were: ettringite: 11.0 ÷ 5.0 mm, thaumasite: 6.0 ÷ 3.5 mm, gypsum: 5.0 ÷ 0.0 mm. Apart from gypsum, in the area of 0.0 ÷ 4.5 mm from the surface of the sample, the bassanite (8) was present. Based on the X-ray patterns of the samples from B-19 specimen, it can be assumed that Friedel’s salt (2) was formed at the expense of carboaluminate (9).

The intensity of the reflections, allowed to conclude, that in the layers located at the greatest distance from the surface in the B-25 sample, portlandite was present in greatest amount. However, this phase could no longer be detected in the 4.0 ÷ 5.0 mm layer. In the most distant layer from the surface (11.0 ÷ 12.0 mm) there were reflections of carboaluminate (9) and calcium aluminate (12), similar to the reference sample B-P. Closer to the surface of the specimen, the reflections of these aluminates disappeared, and a peak of solid solutions formed by Friedel’s salt began to appear. This is especially true for the 9.0 ÷ 10.0 mm layer. Friedel’s salt began to disappear in the layer of 3.0 ÷ 4.0 mm. Ettringite (3) in the 5.0 ÷ 6.0 mm layer was present next to thaumasite (4). However, at a distance of 3.0 to 5.0 mm from the surface, these phases formed a solid solution. The closer to the sample surface, the more the reflection shifts to the right, from the value of 9.09° 2Θ in the layer from 12.0 to 6.0 mm through 9.16° 2Θ in the 5.0 ÷ 6.0 mm layer to 9.26° 2Θ in layers from 3.0 to 5.0 mm. The value of this reflection, at a distance of 2.0 ÷ 3.0 mm, suggests a significant decrease in the content of aluminate ions, so the remaining phase is closer to thaumasite. In the layers ranging from the surface to a depth of 5.0 mm, only gypsum was found (5), which was formed from sulphate phases. The reflections of calcite (6) and vaterite (7) were also more intense closer to the surface. Vaterite was found in layers, distant from the surface up to 3.0 mm. Trace amounts of brownmillite (13) and belite (11) were present in the entire area, which was not found in the 0.0 ÷ 1.0 mm layer.

The results show that in the CEM I hardened cement paste, the hydrated aluminate phases can be found only at a distance of 8 mm from the surface of the samples. This also applies to hydrated aluminate carbonate. On the other hand, in the deeper layers of the slurry, after 25 days of immersion in the NH_4_Cl solution, a hydrogranate (C_3_AH_6_) was found, which was also present in the comparative sample. Belite (dicalcium silicate), an unhydrated clinker phase, was present in different depths in corroded samples, but alite (tricalcium silicate), also an unhydrated clinker phase, was present almost in the same layers as the portlandite. Brownmillerite (C_4_AF) turned out to be the most resistant to the action of ammonium chloride. Calcite was present at the entire depth of the examined specimens, and in the layers closest to the surface, it was accompanied by vaterite, which in the B-4 profile was even detected in the layers located further away from the surface. The sulphate phases formed in the hardened cement paste due to the decrease in pH, in the same order in all the cases. Under these conditions, ettringite was less stable than thaumasite, and thaumasite less than gypsum. In each case, there was an area where all three phases occurred. At the same time, it seems, that only in the case of sample B-4, subjected to the shortest action of the aggressive solution, thaumasite was formed from a solid solution with ettringite. The content of Friedel’s salt gradually decreased towards the surface of the fittings. In samples B-4 and B-19 it was present up to 2 mm, and in B-25—up to 1 mm. In samples taken from B-4, despite its contact with the solution for only 4 days, portlandite was not detectable up to a depth of 2.5 mm. In samples B-19 and B-25, subjected to longer immersion, portlandite was not present in the outer zone of 5.0 mm thick.

In addition to the minerals described above, to complement the analytical results, Table 6 shows also phases that were present in minor amounts.

Microscopic observations

Microscopic observations of the corroded samples confirmed the presence of phases previously detected by diffractometry. Figure 8 shows the solid solution of ettringite with thaumasite in B-4 specimen, at a depth of 2.2 mm. Figure 9 shows the crack in sample B-19 at a depth of at least 4.0 mm, which proves the action of sulphate phases—thaumasite and gypsum, which were present in this place. The process of transforming thaumasite into gypsum, a secondary phase with a smaller volume, disintegrates the matrix. The volume of the phases while maintaining a constant number of moles SO_4_^2−^ is: ettringite → thaumasite → gypsum: 725 → 990 → 222 cm^3^ [6]. In turn, in Figure 10 the mass crystallization of gypsum is shown at a depth of 1.0 mm in sample B-25. Figure 11 shows Friedel’s salt flakes in the center of the photomicrograph taken for sample B-25 at a depth of 6.5 mm. Figure 12 shows spherical and cubical chlorine subsidized calcium carbonate polymorphs and fibrous basic CaCl_2_ in sample B-25 at a depth of 5.7 mm. 

The Figure 13 presents disintegrated, loosened matrix, at a depth of 3.5 mm of the B-25 sample.

Porosity changes

Figure 14 shows significantly changed zone, close to the external sample surface. At the very surface, the pores are filled with corrosion products, mainly: gypsum, calcite, vaterite, bassanite. The porosity assessed using SEM-BSE image analysis confirmed, that with the immersion time, the pores at the edge of the sample become more overgrown with the corrosion products. The highest porosity at the external zone was noticed in the B-4 sample, while the lowest, after longer immersion, in the B-19 and B-25 samples, at the same depth of 2 mm. Due to the deposition of corrosion products in the newly formed pores, the matrix becomes more homogeneous. 

The corrosion products are simple calcium sulfates or carbonates. This calcium probably diffused from the decalcified phases from the deeper layers, and was transported along with the pore water towards the evaporation zone.

Chlorides ingress

The amount of chloride ions at given depths was related to the mass in a respective layer—Figure 15. For more porous layers, chloride ions were determined from the larger surface, perpendicular to the direction of diffusion transport. It can be seen that the maximum chloride content is not located at the very edge of the sample. The content of chloride ions in the hardened cement slurry decreased with the immersion time. This can be explained not only by the diffusion of free chloride ions and the dissociation of ammonium chloride, but also by the dissolution of Friedel’s salt.

Chlorides, porosity, phases—comparison

To show the changes in the phase composition of the corroded, hardened cement paste, at different distances from the surface of the samples, the test results were compiled, based on the intensity of phase reflections on the diffractograms. The content of individual phases in selected layers of the tested samples is shown in Figure 16, depicting the changes in porosity and chloride concentrations. Horizontal lines in the figures correspond to the phase content, porosity and concentration of chloride ions in the reference sample B-P.

Figure 16 shows that the phase transformations of the corroded cement pastes, regardless of the immersion duration, started with the loss of portlandite (reaction (1)), due to the decrease in the pH of the solution in the pores. The AFt (ettringite) phase reaction (2) also lost its stability, which resulted in the formation of Friedel’s salt, which was present up to this point in the solid solutions (reaction (3)). Friedel’s salt began to form in places where excess amount of free chloride ions (in relation to chloroaluminate) was detected (in B-4 at a depth of 8.5 mm, in B-19 at 10.5 mm and in B-25 on 11.5 mm. Subsequently, the depletion of Friedel’s salt was associated with a decrease in the pH to values, where this phase is no longer stable, which was reflected in an increase in the content of free chloride ions. The sharp increase in portlandite content was probably related to the decalcification of the C-S-H phase (reaction (4)). In some zones (depth 8.5 mm in sample B-25, 5.5 mm in B-19 and 3.75 in B-4), a significant amount of calcium ions was detected. Due to such large amounts of calcium, the instability of ettringite, Friedel’s salt and thaumasite (reactions (5) and (6)) began to form. Thaumasite coexisted with ettringite in some areas, forming solid solutions, however, closer to the edge of the sample, thaumasite became a separate phase. The pH range for the stability of thaumasite is narrow, and its dissolution was accompanied by the formation of gypsum and secondary calcite (reaction (7)). However, it is likely that the secondary calcite was formed due to the decomposition of vaterite, which appeared in the greatest amounts in the sample B-4. Moreover, hemihydrate gypsum basanite (S) was detected in sample B-19, which may indicate gypsum dehydration during sampling. The curves of changes in porosity of dry samples (according to the procedure described in [3]) correspond to the amount of CH and FS. A longer aggression time results in the formation of more corrosion products in the edge zone of the sample, as a result of salt transport during drying of the samples.

### 3.2. Modeling

The model part consists in simulations of:-changes in porosity, pH and concentrations of the pore solution components in the model domain after 1, 3, 5, 9, 30 and 40 days from the beginning of the experiment;-changes in the amount of water, mineral phases and pH in the cement matrix, at points located at different distances (0.1 cm, 0.5 cm, 1.1 cm and 2.9 cm) from the beginning of the model domain for 40 days;-changes in the share of selected minerals in the model domain after 1, 3, 5, 9, 30 and 40 days from the beginning of the experiment.

#### 3.2.1. Changes in Porosity, pH and Concentrations of the Pore Solution Components

Upon contact with an aqueous NH_4_Cl solution, the pH of the pore solution decreases inside the sample, the mass of mineral phases decreases and the porosity increases. These phenomena are most intense in the model entry zone (right side of the graphs). The intensity of the changes increases with time. At a distance of about 2.5 cm from the entrance to the model, the changes mentioned, become insignificant—Figure 17.

The Cl^−^, NH_3(aq)_, and H^+^ concentrations were the highest in the model entry zone; they increased and moved in time, with the migration of the solution into the interior of the sample. The SO_4_^2−^ ion presents a different behavior. Its maximum concentration after 1 day of simulation occurred about 0.7 cm from entering the model entry zone and decreased exponentially with the distance from the maximum. In the following days, the maximum concentrations decreased and migrated to the sample interior where the concentrations of sulfate ions were constantly increasing. The changes in Ca^2+^ ion concentrations were similar, but their maximum was located after 1 day, at a depth of about 0.3 cm from the sample edge. Dissolved silica SiO_2(aq)_ was characterized by a decrease in concentrations at the model entry zone, and with time, these concentrations decreased in the entire sample, and 19 days from the beginning of the simulation they were close to zero—Figure 18.

#### 3.2.2. Changes in the Amount of Water and Mineral Phases in the Cement Matrix

Figure 19, Figure 20, Figure 21 and Figure 22 present changes in the amount of water (directly proportional to changes in porosity), mineral phases and pH in the matrix of the hardened cement slurry, occurring during 40 days in example zones, located at different distances: 0.1 cm, 0.5 cm, 1.1 cm and 2.9 cm from the entrance to the model domain. The simulation results proved that the presence of primary and secondary minerals was controlled by the diffusion-penetrating ammonium chloride solution. At a distance of 0.1 cm from the sample—solution contact—Figure 19, a rapid decrease in pH and an increase in the amount of water in the analyzed system are observed in the first 2 days. At the same time, an intensive, complete decomposition of portlandite and the C_1_SH phase occurs, and the degradation of larnite, C_3_S and ettringite (reactions (12)–(16)) is also initiated, the decomposition of which (together with C_1_SH) stimulates the production of small amounts of secondary grossular—a reaction (17).
Ca(OH)_2_ + 2 NH_4_^+^ = Ca^2+^ + 2 H_2_O + 2 NH_3(aq)_(12)
C_1_SH + 4 NH_4_^+^ = 2 Ca^2+^ + 2 H_4_SiO_4(aq)_ + 0.398 H_2_O + 4 NH_3(aq)_(13)
Ca_2_SiO_4_ + 4 NH_4_^+^ = 2 Ca^2+^ + H_4_SiO_4(aq)_ + 4 NH_3(aq)_(14)
C_3_S + 6 NH_4_^+^ = 3 Ca^2+^ + 3 H_2_O + SiO_2(aq)_ + 6 NH_3(aq)_(15)
Ca_6_Al_2_(SO_4_)_3_(OH)_12_·26H_2_O + 12 NH_4_^+^ = 6 Ca^2+^ + 38 H_2_O + 2 Al^3+^ + 3 SO_4_^2+^ + 12 NH_3(aq)_
(16)
12 NH_3(aq)_ + Ca_6_Al_2_(SO_4_)_3_(OH)_12_·26H_2_O + 3 C_1_SH = Ca_3_Al_2_Si_3_O_12_ + 6Ca^2+^ + 29 H_2_O + 3 SO_4_^2−^ + 12 NH_4_^+^(17)

Each of the mentioned reactions leads to the production of ammonia, or to local changes between NH_4_^+^/NH_3(aq)_, and often, except for the reaction (14), significant amounts of water. It should be mentioned here that ammonia was detected organoleptically during laboratory experiments. After about 3 days, a secondary phase—okenite was formed, the decomposition of which begins on the 7th day of the modeled process:CaSi_2_O_4_(OH)_2_·H_2_O + 2NH_4_^+^ = Ca^2+^ + 3H_2_O + 2SiO_2_ + 2NH_3(aq)_.(18)

After about 15 days from the beginning of contact of the sample with the external solution, the changes in the amount of the main mineral phases were negligible. In the analyzed period, the pH decreased from 12.4 to 8.4, while the amount of water increased from 0.0022 mol to 0.0077 mol.

At a distance of 0.5 cm from the contact of the sample with the solution—Figure 20, on the first day a rapid decrease in pH and an increase in the amount of water in the analyzed system were observed.

At the same time, there was an intensive, complete decomposition of the C_1_SH phase. The degradation of portlandite, larnite, C_3_S and ettringite was initiated, reactions (12)–(17). The decomposition of ettringite, together with C_1_SH, stimulated the production of small amounts of secondary grossular—reaction (17). Grossular decomposed from day 2, which enables the formation of the C_2_SH_a_ phase (19):4 H^+^ + 0.6667 Ca_3_Al_2_Si_3_O_12_ + H_2_O = C_2_SH_a_ + H_4_SiO_4(aq)_ + 1.333 Al^3+^(19)

C_3_S (20) also takes part in this process:0.6667 C_3_S + H_4_SiO_4(aq)_ = C_2_SH_a_ + H_2_O + 0.6667 SiO_2(aq)_(20)

The degradation of C_2_SH_a_ takes place from day 10 to 12, when this phase is completely depleted. Until about day 18, the decomposition of portlandite continues; after this time, the changes in the amount of the main mineral phases are negligible. In the analyzed period of 40-day aggression, the pH drops from 12.4 to 9.4, while the amount of water increases from 0.0022 mol to 0.0068 mol.

At a distance of 1.1 cm from the contact surface of the sample with the solution—Figure 21, on the first day of the simulation, there was a rapid decrease in pH and the amount of water, with simultaneous decomposition of portlandite, C_1_SH and C_3_S. The crystallization of the secondary C_2_SH_a_, that began on the second day, was mainly due to the degradation of C_1_SH:C_1_SH + 0.602 H_2_O = C_2_SH_a_ + H_4_SiO_4(aq)_.(21)

At the same time, recrystallization of C_3_S takes place, and the increase in C_2_SH_a_ is suppressed.

From the 21st day the formation of the calcium hydroxichloride (Ca:H_2_O) took place: Ca^2+^ + 3 H_2_O + Cl^−^ = CaCl(OH)·2H_2_O + H^+^.(22)

In the analyzed period, the pH decreased from 12.4 to 11.8, while the amount of water increased from 0.0022 mol to 0.0024 mol.

At a distance of 2.9 cm from the contact of the sample with the solution—Figure 22, on the first day of the simulation there was a rapid decrease in pH and the amount of water, with simultaneous decomposition of portlandite, C_1_SH and C_3_S.

The crystallization of the secondary C_2_SHa, that begins on the second day is mainly due to the decomposition of C_1_SH—reaction (21).

At the same time, the recrystallization of C_3_S, to the initial amount, took place and the increase in C_2_SHa amount was suppressed. From the 10th day, no changes in the amount of the main phases were observed. In the analyzed period, the pH decreased from 12.4 to 11.8, while the amount of water increased from 0.0022 mol to 0.0024 mol.

#### 3.2.3. Changes in the Share of Selected Minerals

Figure 23 shows the profiles of changes in the share of selected minerals, in the model domain, at different times, from the beginning of the experiment. In the modeling period, in the entire sample, decreases in the shares of the primary components were observed: portlandite, larnite, and especially C_1_SH and C_3_S. 

Changes in the ettringite, C_3_S and Friedel’s salt are varied: they decrease in the edge part of the sample, the enrichment occurs in some distance from the sample entrance, and then another decrease occurred in the zone located approximately in the middle of the sample.

Among the hydrates, the most intense increases are recorded for C_2_SHa, which already on the first day of the simulated contact of the NH_4_Cl solution with the sample, reached almost 20%, in the input zone. In the following days, this share systematically grows and reaches a uniform value of about 31% inside the sample. As time passes, a C_2_SHa-free zone develops from the entry side of the model. Generally, in the subsequent stages of modeling, it is observed, the expansion of the zones in which the primary phases have completely decomposed. They can be replaced by episodic secondary phases, such as, for example, grossular, gyrolite, heulandite, yennite, tobermorite.

#### 3.2.4. Summary of the Model Part

On the basis of the simulations, the presence of individual mineral phases in subsequent cells of the model, corresponding to the zones located at different distances from the entrance to the model, was identified—Table 7. 

## 4. Discussion 

The outcomes of the laboratory tests are essentially in line with the modeling results. Some differences result mainly from the fact that geochemical simulations, contrary to laboratory experiments, do not take into account the three-month seasoning in limewater (which would not be the case in the practical application of cements). Moreover, the cements after laboratory tests were dried in air conditions, samples were collected using the abrasive method, and then their composition was tested. On the other hand, the simulation results allow to track changes taking place in the samples and assess their composition at any time, without interrupting the virtual contact of the cement with the solution.

Laboratory tests revealed significant amounts of gypsum detected on the very edge of the samples, while geochemical modeling did not indicate its crystallization. The same is true for thaumasite—a transition phase between ettringite and gypsum, which is formed under specific conditions. 

Most probably, the analyzed samples, which were subjected to laboratory experiments and drying, were enriched with carbonate phases (vaterite, calcite) and gypsum, that precipitated from evaporating pore solution. The saturation indices (SI) calculated for these phases in the geochemical models, did not indicate the possibility of their crystallization, under the simulated conditions of continuous contact of the sample with NH_4_Cl solution.

The model is able to indicate non-crystalline phases which cannot be detected by X-ray diffraction, such as silicic acid gel or unstable silica-β.

The differences in the results between the model and the experiment may result from the fact that in laboratory conditions, there was a diffusion exchange of ions between the cement and the surrounding NH_4_Cl solution, in the case of the model, diffusion transport takes place only to the inside of the cement sample.

Changes in the porosity in the model are characterized by the presence of a straight line (middle part of the graph), sometimes horizontal, within a certain depth range (Figure 17), similarly to other authors dealing with modeling the decalcification process [24,25,29,30,33], which is related to the loss of portlandite. The porosity then increased towards the edge of the sample as other calcium phases were dissolved. This relationship could not be observed in the samples subjected to laboratory tests (Figure 9 and Figure 14), or the increase in porosity was slight (Figure 16), because the pores were getting clogged with corrosion products, in particular calcium carbonates and gypsum, mainly in the outer layer.

## 5. Conclusions

The test results demonstrated a detrimental impact of the aqueous solution of ammonium chloride salt on the hardened cement paste, expressed by significant changes in the phase composition and increased porosity.

The contact of the aqueous NH_4_Cl solution with the sample causes a decrease in the pH of the pore solution, and an increase in porosity due to the degradation of the primary mineral phases: mainly portlandite and C_1_SH, and to a lesser extent also larnite, C_3_S and ettringite. Each of the above-mentioned reactions leads to the production of significant amounts of water as well as ammonia, which was detected organoleptically during laboratory experiments.

Precipitation of secondary phases, such as grossular, C_2_SHa, okenite and calcium hydroxichloride, does not balance the volume of decomposing phases, hence the increase in porosity results, which is most intense in the contact zone. The share of C_2_SHa increased most intensively, and after 19 days of simulation it reached a constant value of about 31%.

With time (in the subsequent stages of modeling), the zones of complete decomposition of the primary phases penetrated deeper into the sample. The primary phases could also be replaced by episodic secondary minerals, such as, for example, grossular, gyrolite, heulandite, jennite and tobermorite.

The results of laboratory tests and model analyzes can perfectly complement each other and explain the phenomena of destruction occurring in the cement. With laboratory tests, we are not able to determine the actual porosity caused by chemical destruction, if corrosion products crystallize in the pores, and this takes place when free water is drained from the material while drying. The model is able to show this porosity and the resulting material decay.

By comparing the analyzes of degraded dry and wet material, it is possible to describe the expected destruction of the reinforced concrete structure elements above the exemplary coke sewage table, in coke wastewater tanks, which may occasionally be empty. A similar case may also apply to reinforce concrete columns partially submerged in sea water, supporting coastal buildings, or to structures in agricultural land with a high groundwater table.

Further work on the issues presented is necessary. We intend to check what crystalline phases will remain or precipitate in the degraded continuous medium, in the event of the removal of diffusing ions, or the dehydration process.

## Figures and Tables

**Figure 1 materials-15-02930-f001:**
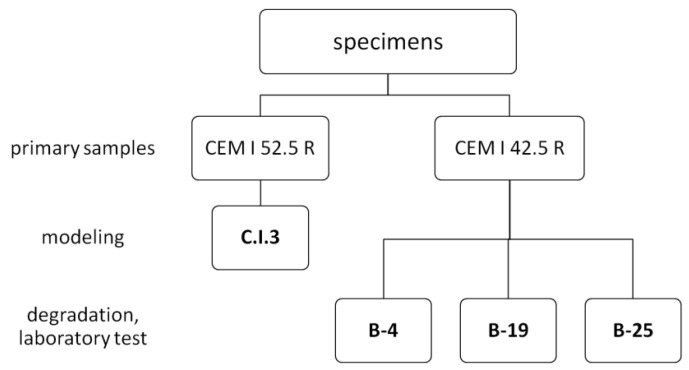
Specification of tested samples.

**Figure 2 materials-15-02930-f002:**
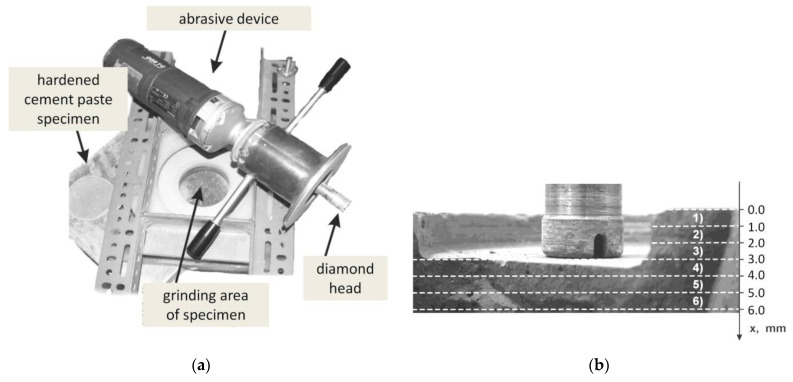
Sampling of powdered samples: (**a**) Workplace with abrasive device; (**b**) Diamond head and exemplary layering.

**Figure 3 materials-15-02930-f003:**
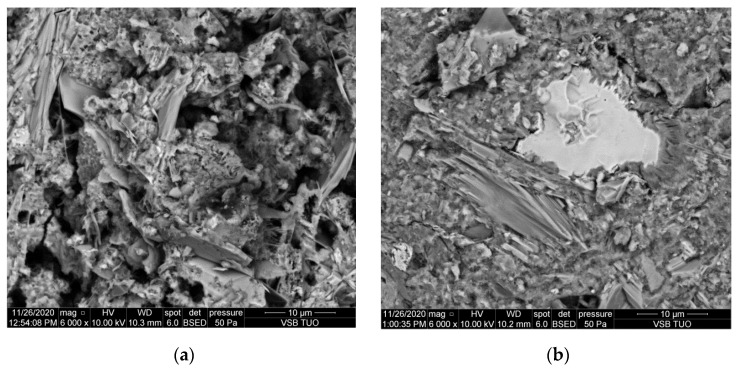
SEM images of the microareas of C.I.3 specimen: (**a**) Needle-like crystals of ettringite; (**b**) Platty crystals of portlandite.

**Figure 4 materials-15-02930-f004:**
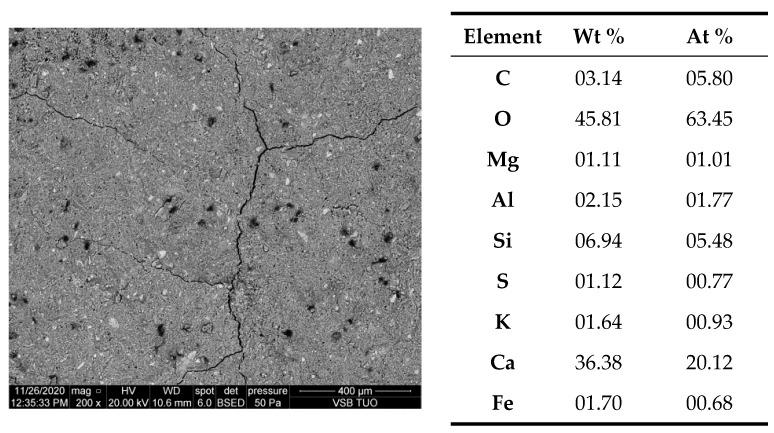
SEM image of the microarea of C.I.3 specimen with its elemental composition, by weight—(Wt) and atomic proportions—(At).

**Figure 5 materials-15-02930-f005:**
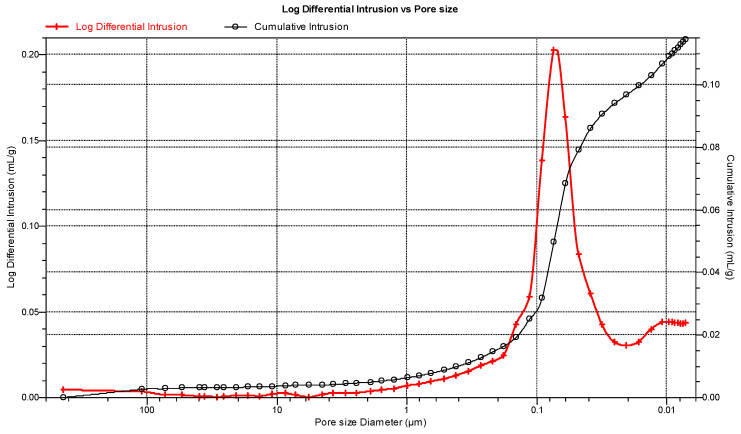
Pore size distribution in C.I.3 specimen.

**Figure 6 materials-15-02930-f006:**
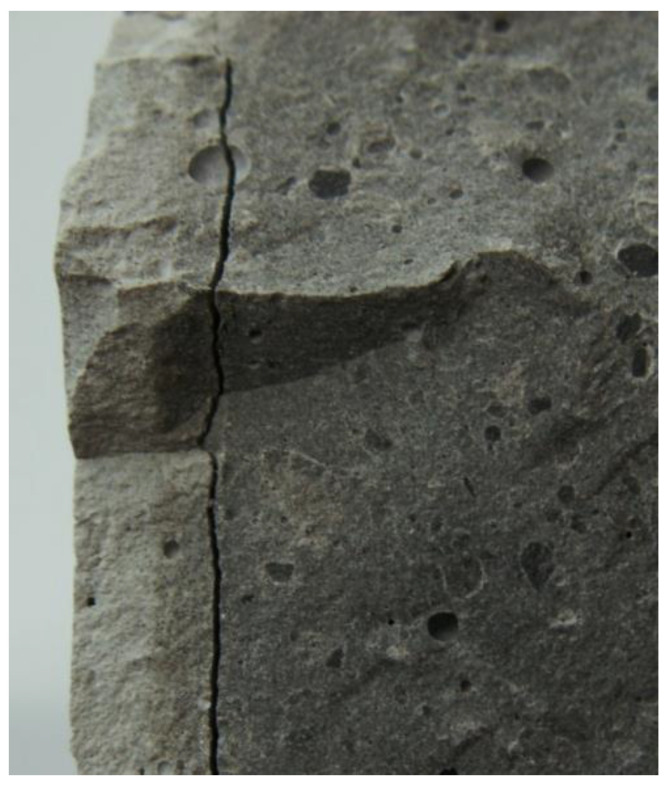
Fracture with a visible crack at the border of altered zone at a depth of 5.5–6.0 mm: B-25 specimen.

**Figure 7 materials-15-02930-f007:**
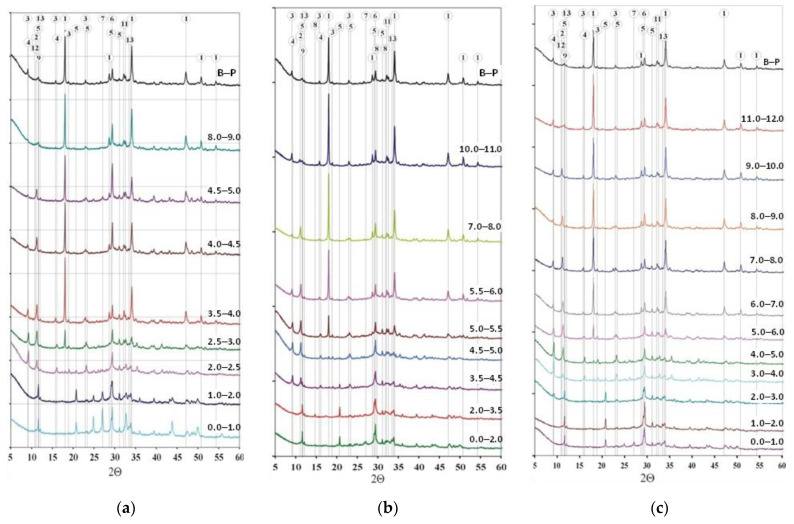
XRD patterns of (**a**) B-4, (**b**) B-19 and (**c**) B-25 specimen. Explanations in the text.

**Figure 8 materials-15-02930-f008:**
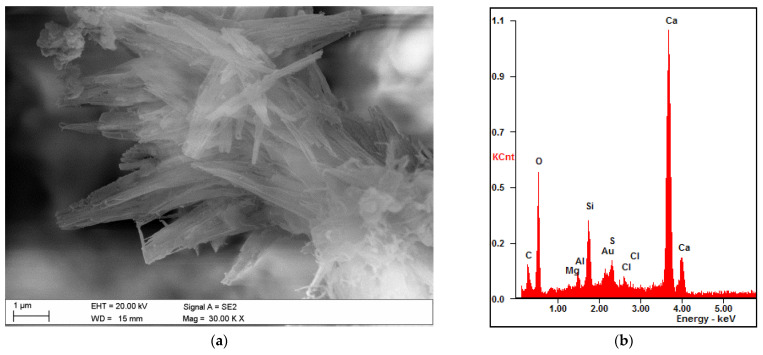
Solid solution of ettringite with thaumasite in B-4 specimen at a depth of 2.2 mm: (**a**) SEM image; (**b**) Chemical composition.

**Figure 9 materials-15-02930-f009:**
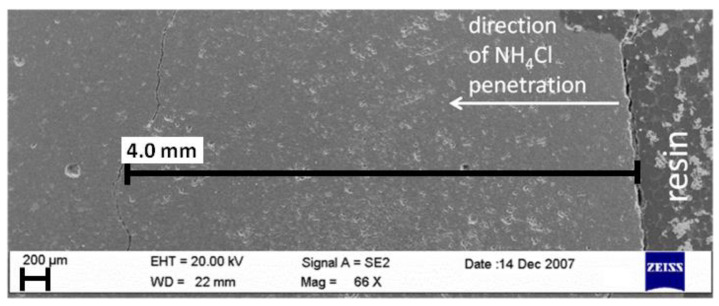
A micrograph of the polished section of sample B-19 with marked crack at a distance of at least 4.0 mm from the contact surface with the aggressive solution where the thaumasite started to dissolve.

**Figure 10 materials-15-02930-f010:**
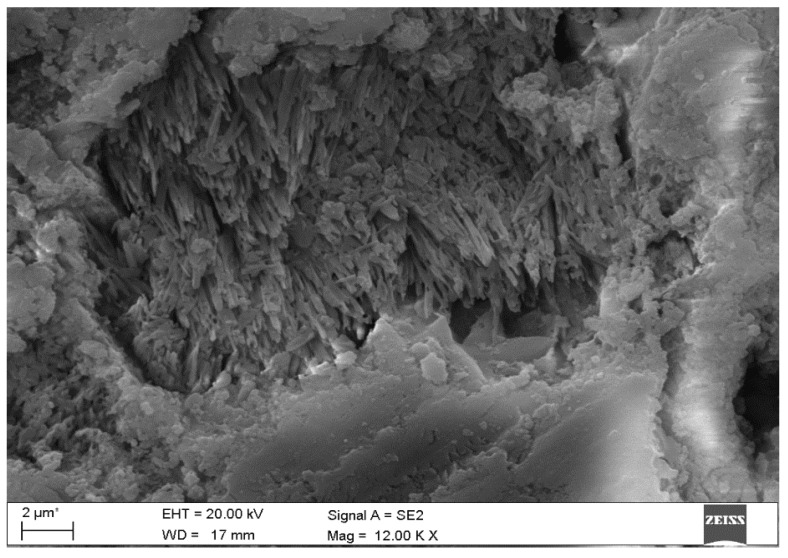
Filling the pore with gypsum at a depth of 1.0 mm in sample B-25.

**Figure 11 materials-15-02930-f011:**
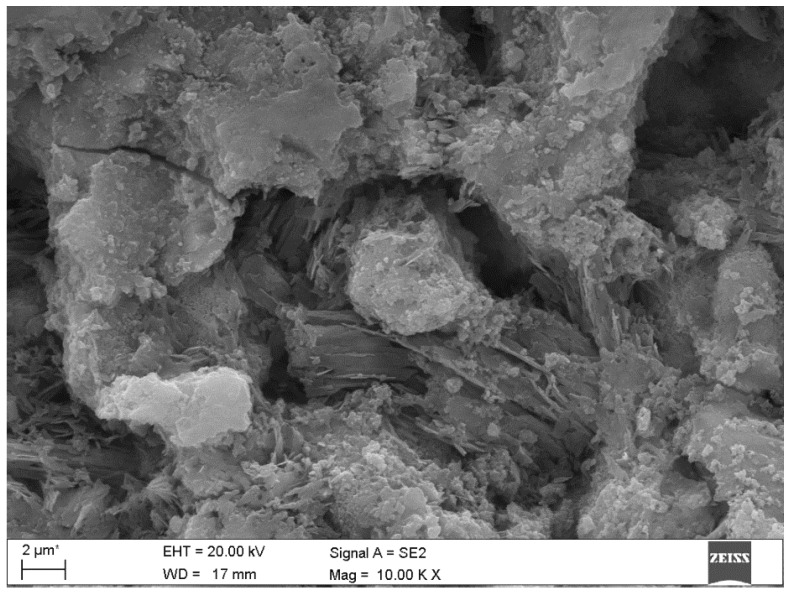
Friedel’s salt flake phase in the center of the photomicrograph taken for sample B-25 at a depth of 6.5 mm.

**Figure 12 materials-15-02930-f012:**
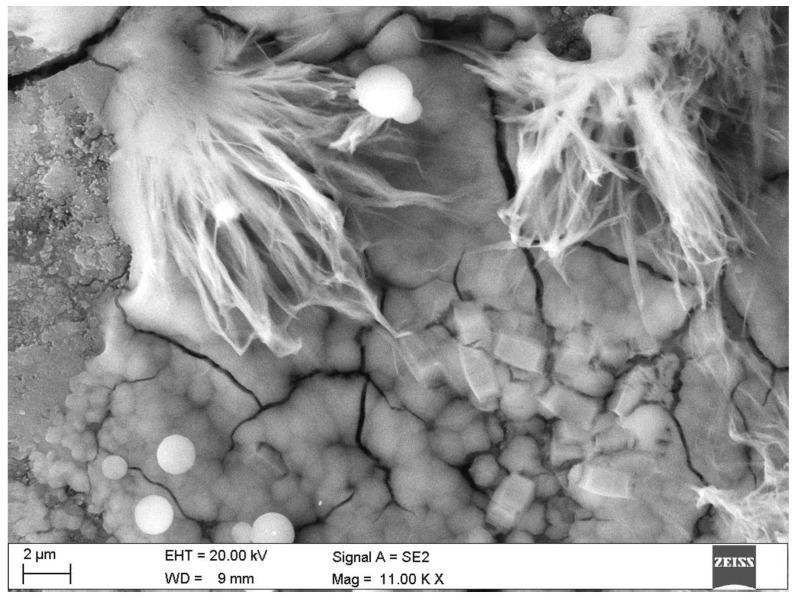
Chlorine subsidized calcium carbonate polymorphs: cubes and basic CaCl_2—_fibers. Sample B-25, depth 5.7 mm.

**Figure 13 materials-15-02930-f013:**
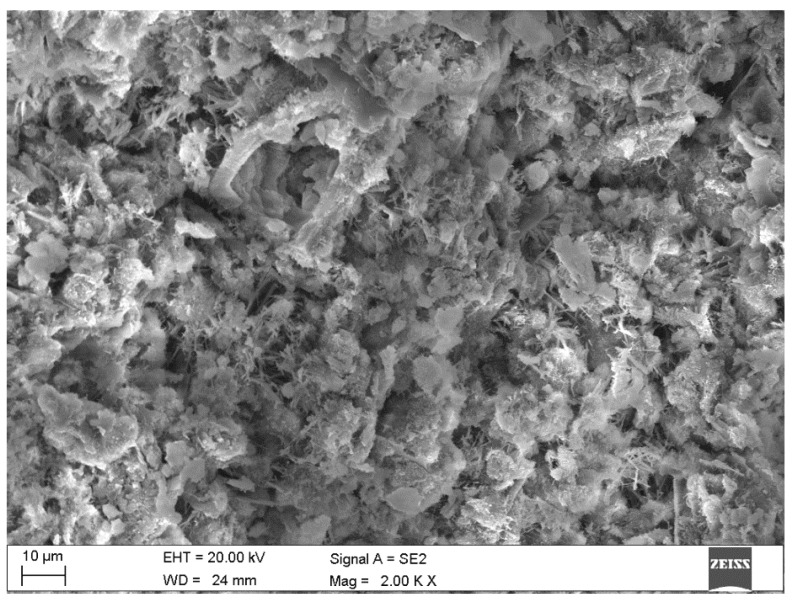
Disintegrated, loosened, non-cohesive sample matrix of B-25 at a depth of 3.5 mm.

**Figure 14 materials-15-02930-f014:**
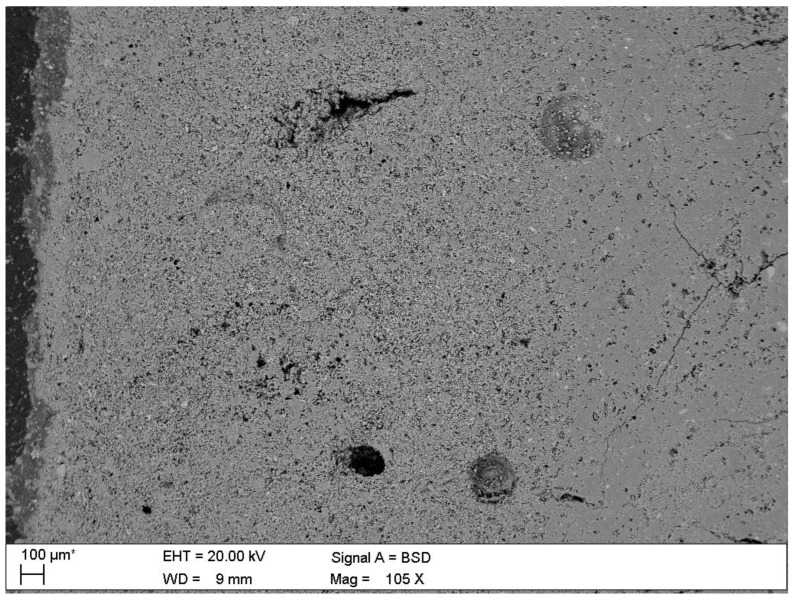
Increased porosity at the edge of sample B-4 to about 2400 μm and increased density of the specimen on the very surface with a thickness of up to 100 μm (external edge of the sample is located at the left side of the image).

**Figure 15 materials-15-02930-f015:**
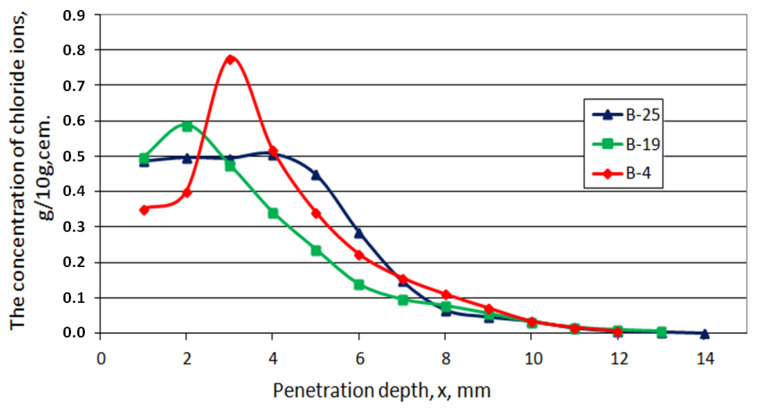
Concentration of free chloride ions in series B of hardened cement paste, depending on the transport way and immersion time.

**Figure 16 materials-15-02930-f016:**
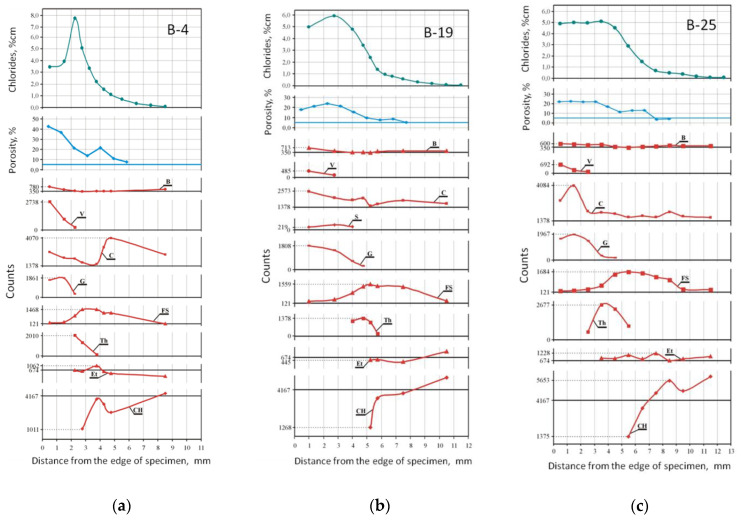
Changes in the concentration of free chloride ions (% of mass of cement), changes in porosity (%), changes in the phase content depending on the distance from the specimen’s surface in (**a**) B-4, (**b**) B-19 and (**c**) B-25, abbreviated names of phases—as in Table 6 and in Appendix A.

**Figure 17 materials-15-02930-f017:**
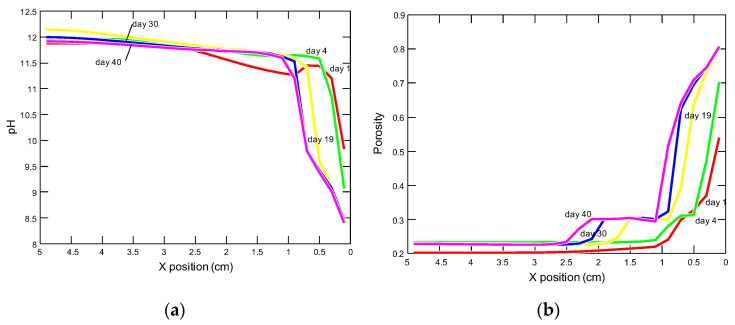
Profiles of changes in (**a**) pH and (**b**) porosity in the model domain after 1, 4, 19, 30 and 40 days from the beginning of the experiment.

**Figure 18 materials-15-02930-f018:**
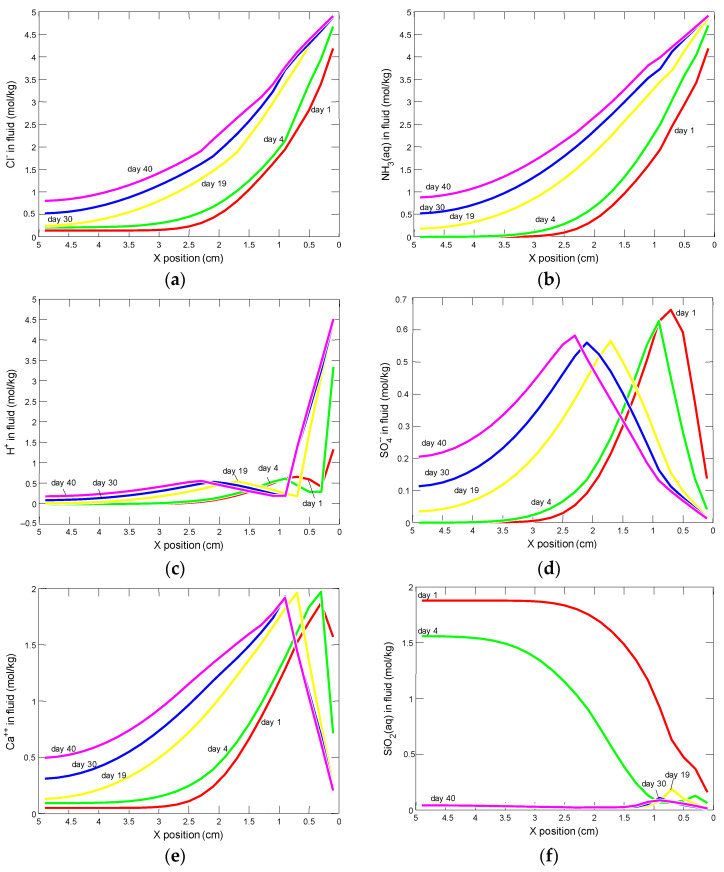
Profiles of changes in the concentrations of the components of the pore solution in the model domain, after 1, 4, 19, 30 and 40 days from the beginning of the experiment, for: (**a**) chlorides, (**b**) ammonia, (**c**) hydrogen, (**d**) sulphates, (**e**) calcium and (**f**) silica.

**Figure 19 materials-15-02930-f019:**
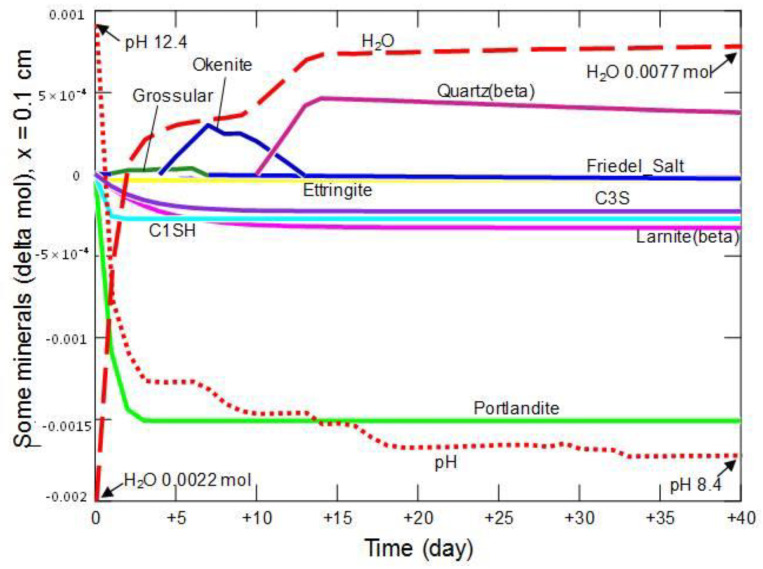
Changes in the amount of water, mineral phases and pH in the cement matrix at a distance of 0.1 cm from the beginning of the model domain, in conditions of ammonium chloride penetration.

**Figure 20 materials-15-02930-f020:**
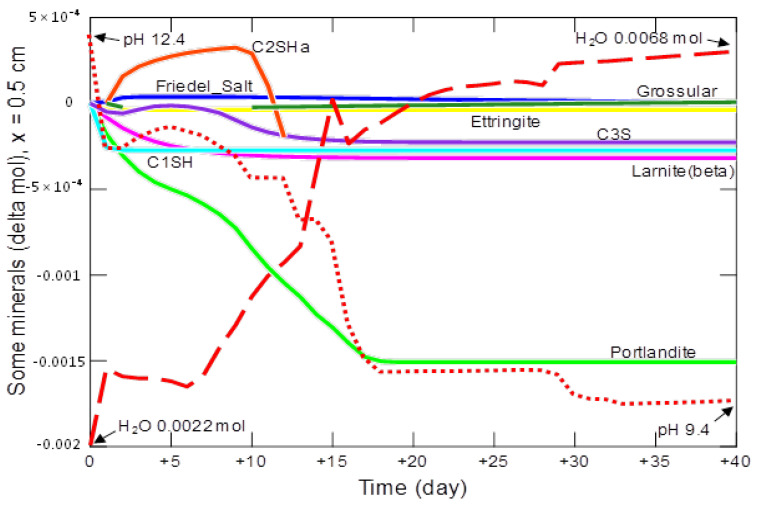
Changes in the amount of water, mineral phases and pH in the cement matrix at a distance of 0.5 cm, from the beginning of the model domain, in conditions of ammonium chloride penetration.

**Figure 21 materials-15-02930-f021:**
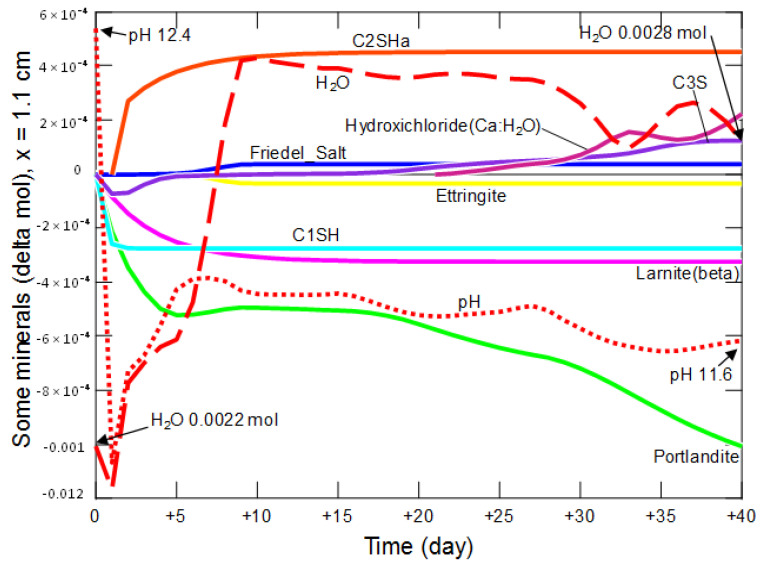
Changes in the amount of water, mineral phases and pH in the cement matrix at a distance of 1.1 cm from the beginning of the model domain in conditions of ammonium chloride penetration.

**Figure 22 materials-15-02930-f022:**
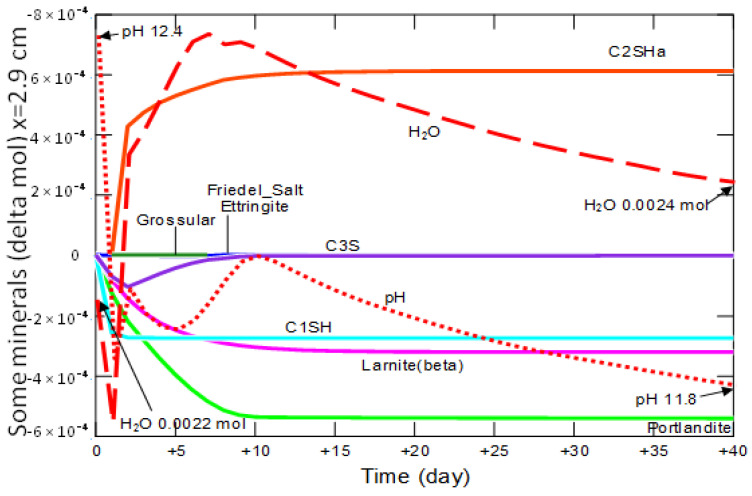
Changes in the amount of water, mineral phases and pH, in the cement matrix at a distance of 2.9 cm from the beginning of the model domain in conditions of ammonium chloride penetration.

**Figure 23 materials-15-02930-f023:**
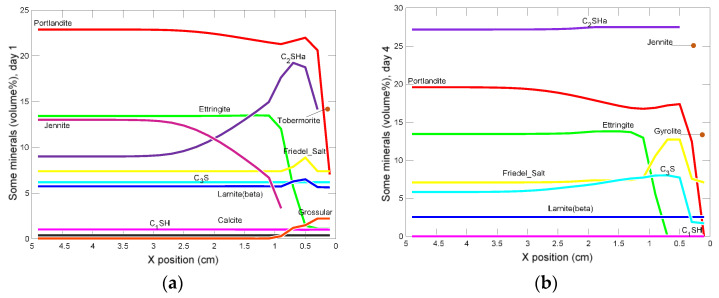
Profiles of changes in selected minerals in the domain of the model, in (**a**) 1, (**b**) 4, (**c**) 19, and (**d**) 30 days from the beginning of the experiment.

**Table 1 materials-15-02930-t001:** Calculated pH of water in equilibrium with the solid phase, at 20 °C [1].

Phase	Portlandite CH	Calcium Aluminate C_3_AH_6_	Carboaluminate C_3_A·CĈ·H_11_	Ettringite C_6_AŜ_3_·H_32_	Calcite CĈ	Thaumasite C_3_SĈŜ·H_15_	Gypsum CŜH_2_
**pH**	12.54	11.64	11.37	11.00	9.96	8.19	7.07

**Table 2 materials-15-02930-t002:** Chemical composition of CEM I 42.5 R and CEM I 52.5 R cement.

	Oxide Mass %							
	CaO	SiO_2_	Al_2_O_3_	Fe_2_O_3_	SO_3_	K_2_O	Na_2_O	MgO
CEM I 42.5 R	62.63	19.03	5.60	2.89	3.14	0.98	0.16	n.d.
CEM I 52.5 R	63.26	19.65	5.30	2.85	3.14	0.96	0.21	1.55

n.d.—not detected.

**Table 3 materials-15-02930-t003:** Kinetic reaction parameters for the temperature of 25 °C and specific surfaces of mineral grains adopted for modeling.

Mineral	Dissolution Rate—*k_25_*		Specific Surface Area SSA—*A_n_*		Source
	Acid Mechanism	Neutral Mechanism			
	(mol/m^2^·s^−1^)	(mol/m^2^·s^−1^)	In Model (cm^2^/g)	In Literature (m^2^/g)	
Ettringite	1.14 × 10^−12^ *	1.14 × 10^−12^ *	20,000 *	2 *	(*) Labus, Wertz, 2017 [36] (after Baur et al., 2004) [37]
Calcite	5.00 × 10^−1^ *	1.55 × 10^−6^ *	260	0.026 **	(*) Palandri, Kharaka, 2004; [38] (**) Labus, Wertz, 2017 [36]
Portlandite	8.04 × 10^−4^ *	2.18 × 10^−8^ *	20,000	2 *	(*) Labus, Wertz, 2017 [36] (after Gali et al., 2001) [39]
Hydrocalumite Friedels Salt Ca_4_Al_2_Cl_2_O_6_:10H_2_O	5.89 × 10^−10^ *	5.89 × 10^−10^ *	5000	6.1; 7.1; 9.4; 13.7, 35.4 *	(*) Marty et al., 2017 [40]
Quartz	7.76 × 10^−12^ *	1.02 × 10^−14^ *	157.3	0.01573 **	(*) Palandri, Kharaka, 2004; [38] (**) Labus, Wertz, 2017 [36]
C_2_S-β Larnite	1.0 × 10^−6^ *	1.0 × 10^−6^ *	20,000	2.93 **	(*) Brand et al., 2019 [41] (**) Nicoleau et al., 2013 [43]
C_1_SH	5.94 × 10^−8^ *	1.60 × 10^−18^ *	20,000	2 *	(*) Labus, Wertz, 2017 [36] (after Schweizer 1999) [42]
C_3_S	5.94 × 10^−8^ *	1.60 × 10^−18^ *	20,000 *	-	(*) assumed as for C1SH

**Table 4 materials-15-02930-t004:** Compositions of the pore solution and the NH_4_ solution used in modeling.

Parameter	Unit	Pore Solution	NH_4_Cl Solution
pH	[–]	11.76	4.34
Ca^2+^	[mol/kg]	6.7 × 10^−3^	1.0 × 10^−5^
Mg	[mol/kg]	1.0 × 10^−6^	1.0 × 10^−4^
Cl^−^	[mol/kg]	0.444	7.20
NH_4_^+^	[mol/kg]	1.0 × 10^−6^	7.20
K^+^	[mol/kg]	1.0 × 10^−6^	1.0 × 10^−6^
Na^+^	[mol/kg]	0.426	1.0 × 10^−3^
SO_4_^2−^	[mol/kg]	6.7 × 10^−3^	1.0 × 10^−6^
HCO_3_^−^	[mol/kg]	1.0 × 10^−6^	1.0 × 10^−4^
H_4_SiO_4(aq)_	[mol/kg]	1.0 × 10^−6^	1.0 × 10^−6^
Al^3+^	[mol/kg]	1.0 × 10^−6^	1.0 × 10^−6^
Fe	[mol/kg]	1.0 × 10^−6^	1.0 × 10^−6^

**Table 5 materials-15-02930-t005:** Phase composition of hardened cement paste—C.I.3. specimen.

Phase	Content % mass ± 3σ	R_exp_	R_wp_	Content % vol
Ettringite	15.71 ± 0.78	3.23	5.98	13.44
Calcite	0.70 ± 0.36			0.39
Portlandite	37.0 ± 1.60			24.96
Hydrocalumite (Friedel’s salt)	10.08 ± 0.67			7.50
Silica	0.91 ± 0.18			0.52
Dicalcium silicate beta	18.4 ± 1.10			8.33
Tricalcium silicate monocl.	17.2 ± 1.50			8.33
Amorphous calcium silicate hydrate C_1_SH	approx. 20			16.67

**Table 6 materials-15-02930-t006:** Occurrence of crystalline phases in layers of corroded and reference samples.

Symbol in XRD Set of Patterns	Specimen Phase	B-4								B-19								B-25											B-P
		Layer [mm]																											
		0.0–1.0	1.0–2.0	2.0–2.5	2.5–3.0	3.5–4.0	4.0–4.5	4.5–5.0	8.0–9.0	0.0–2.0	2.0–3.5	3.5–4.5	4.5–5.0	5.0–5.5	5.5–6.0	7.0–8.0	10.0–11.0	0.0–1.0	1.0–2.0	2.0–3.0	3.0–4.0	4.0–5.0	5.0–6.0	6.0–7.0	7.0–8.0	8.0–9.0	9.0–10.0	11.0–12.0	
1	Potlandite (CH)				+	+	+	+	+					+	+	+	+						+	+	+	+	+	+	+
2	Friedel’s salt (FS)	*	*	+	+	+	+	+	+	*	+	+		+	+	+	+		+	+	+	+	+	+	+	+	/	*	*
3	Ettringite (Et)			* /	x	+	+	+	+				* /	+	+	+	+				* /	+	+	+	+	+	+	+	+
4	Thaumasite (Th)			+	x	+						+	+	+	*					+	+	+	+						
5	Gypsum (G)	+	+	+						+	+	+	+					+	+	+	+	+							
6	Calcite (C)	+	+	+	+	+	+	+	+	+	+	+	+	+	+	+	+	+	+	+	+	+	+	+	+	+	+	+	+
7	Vaterite (V)	+	+	+				+		+	+							+	+	+									
8	Bassanite (S)									+	+	+																	
9	Carboaluminate								+						+	+	+											+	
11	Belite (larnite)			+	+	+	+	+	+	+	+	+	+	+	+	+	+		+	+			+	+	+	+		+	+
12	Calcium aluminate							+	+																		+	+	+
13	Brownmillerite (B)	+	+	+	+	+	+	+	+	+	+	+	+	+	+	+	+	+	+	+	+	+	+	+	+	+	+	+	+
-	Alite (hatrurit)				+	+	+	+	+				+	+	+	+	+						+	+	+	+	+	+	+
-	Calcium aluminate						+	+	+															+			+		
-	Katoite									+	+	+												+			+		
-	Gismondine										+	+							+	+								+	
-	Thenardite			+								+	+							+									

Explanations: (+)—evidently present, (*)—present in traces, (/)—present in solid solution.

**Table 7 materials-15-02930-t007:** Occurrence of mineral phases in successive cells of the model, identified on the basis of simulation (B-P—reference sample), (**+** in table means that phase was present in a depth).

The Day of Modeling Phase	Day 4								Day 19								Day 25								
	Cell Position from the End of the Model Domain [mm]																								
	1.0	3.0	5.0	7.0	9.0	11.0	13.0	29.0	1.0	3.0	5.0	7.0	9.0	11.0	13.0	29.0	1.0	3.0	5.0	7.0	9.0	11.0	13.0	29.0	B-P
Portlandite		+	+	+	+	+	+	+				+	+	+	+	+				+	+	+	+	+	+
Friedel’s Salt	+	+	+	+	+	+	+	+	+	+	+	+	+	+	+	+	+	+	+	+	+	+	+	+	+
Ettringite				+	+	+	+	+								+								+	+
Calcite	+	+	+	+	+	+	+	+		+	+	+	+	+	+	+		+	+	+	+	+	+	+	+
Aragonite		+																							
C_1_SH																									+
Larnite (β)	+	+	+	+	+	+	+	+	+	+	+	+	+	+	+	+									+
C_3_S	+	+	+	+	+	+	+	+			+	+	+	+	+	+			+	+	+	+	+	+	+
C_2_SH_a_			+	+	+	+	+	+				+	+	+	+	+					+	+	+	+	
Quartz (α)	+	+	+	+	+	+	+	+	+	+	+	+	+	+	+	+	+	+	+	+	+	+	+	+	+
Quartz (β)									+								+								
Grossular	+	+						+		+	+							+	+	+					
Jennite		+																		+					
Tobermorite											+								+						
Diopside	+									+								+							
Gyrolite	+									+	+							+	+						
Hydroxichloride			+										+								+	+			
Wollastonite		+																		+					
Okenite	+																								
Heulandite									+								+								

## Data Availability

Not applicable.

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
