# Peer review of "Laboratory Test and Geochemical Modeling of Cement Paste Degradation, in Contact with Ammonium Chloride Solution"

_materials, 2022, doi:10.3390/ma15082930_

Round 1

Reviewer 1 Report

This paper investigates the ions transport and reaction in concrete. The authors employed multiple effective test methods to reveal the transport-reaction mechanism and get some interesting findings. Overall, the manuscript is well-organized and easy to follow. Minor corrections in the paper to be incorporated.

  1. Please provide more in-depth discussion of the phase composition evolution due to ammonium chloride attack.
  2. Line219-228:Variable symbols are best written in italics
  3. Line35:laboratory and field works in https://doi.org/10.3390/app11020888 indicate that aggressive medium have an significant effect on concrete deterioration.
  4. Please proofread the paper to correct grammatical errors.
  5. Please verify the format of the bibliography (https://doi.org, or doi).

Reviewer 2 Report

A lot of effort has been invested in these analyzes, just try to connect at least some of the properties with the properties defined by standards of the finished product used in construction.

Reviewer 3 Report

Тhe article presents the test results.
They showed the harmful effect of an aqueous solution of ammonium chloride salt on the hardened cement paste, which is expressed in significant changes in the phase composition and an increase in porosity.
This work is very well developed by the authors. There are some problems that can be corrected.

1.Some Keywords are poorly cited in the text, for example, Keyword "durability" is not present in the text even once. Authors must correct all Keywords.
2. Table 1 needs to be rebuilt, position parameters in the vertical direction are not readable.
3. At the end of the first chapter, authors are encouraged to briefly describe the goals and work plan, divided into paragraphs.
4. The authors are recommended to reduce formula (9) to the form indicated in [36], [38].
Why do the authors not describe the parameters p and q are empirical and dimensionless, presented in similar formulas on the example of works [36] and [38]?
Three parameters are used in formula (10): kN25, kH25, kOH25 - where are their values and values of the corresponding components used in expression (10) described?
5. In Table 3, authors are encouraged to indicate article index numbers along with the Source list.
6. In Table 3, the expression mol/m 2 s needs to be corrected to the correct
7. In Table 3, it is necessary to give a decoding in the form of variables of all parameters used in the formulas.
8. What values does the parameter E - activation energy (J / mol) (specified in line 226) take? These parameters should be presented in Table 3.
9. The authors did not present the results of calculating the Reaction kinetics parameters (line 271). Lines 271 272 do not indicate the number of the literature source (from the literature....?)
10. Description of the parameters specified in formula (11) and their description below (lines 254 255) must correspond
11. Line 314 contains the parameter tortuosity - 1095.86, units of measurement are not specified.
12. Is it possible to combine the graphs in Figure 16 for comparison and give comparative conclusions based on the results of the studies?
13. In Fig. 17, the inscriptions should be placed in the form of a side legend.
14. What do the "*" and "/" icons in Table 6 mean?
15. In figure 4, the middle figure, all labels that do not relate to the graph in the figure should be removed. on the right side there is a table possibly related to the figure in the middle. If so, then the designations presented in the figure and in the table must correspond. Correction required.
16. Give an explanation of the parameters Wt % At % indicated in Figure 4 in the table.
17. Missing sign under figure 7.
18. On the chart of Figure 8 on the right side of the Energy picture, there may be repeating symbols.
In Figure 8, the distance 4.0 mm is not clearly shown in green.
19. Is it possible to rotate the x-axis position(cm) in Figure 17 so that 0 starts from the left?
20. Is it possible to rearrange the plots of Figure 18 (concentrations of the components of the pore solution) for each of the individual components to compare them?
In these graphs, the dynamics of the change in the components is not visible.
21. From the graphs in Figure 17, it can be seen that at a depth level after 1 cm, the pH drops sharply and the level of porosity sharply increases.
Is there a relationship between these parameters? Could the authors of the article explain this fact?
22. Are formulas (19), (20) and (21) written correctly?
23. I draw your attention to the fact that the authors need to correct the units of measurement in the unified metric system throughout the text, for example, change cm to mm.
24. It is recommended to remove Ph. D Thesis from the References list. (N 1, N42) and replace with current sources from the works of these authors.
25. It is recommended to replace the source of literature 38 with the cited source of the same authors

Round 2

Reviewer 2 Report

Thanks for the clarifications. My suggestion was to include the names of the norms according to which the tests were performed in the text so that the possible comparability of the test results is visible. The same applies to the requirements for the characteristics of the finished product. I have no further objections.

Author Response

Thanks for your valuable suggestions, if it is possible, we will include in the text the names of the standards according to which the research was carried out, however, they were followed according to scientific needs. Standards are not always appropriate. Not all are applicable, however, for example, we must test a cementless, slag, binder using cement standards. Undoubtedly, there is a need to develop new standards, especially for geopolymeric binders.

Thank you. All the best!